

# Differential responses of the seed germination of three functional groups to low temperature and darkness in a typical steppe, Northern China

Mengzhou Liu[1,*], Ning Qiao[2,*], Bing Zhang[2], Fengying Liu[3], Yuan Miao[2], Ji Chen[4,5,6], Yanfeng Sun[7], Peng Wang[8] and Dong Wang[2]

[1] College of Geography and Environmental Science, Henan University, Kaifeng, China
[2] International Joint Research Laboratory of Global Change Ecology, School of Life Sciences, Henan University, Kaifeng, China
[3] Institute of Microbial Engineering, Laboratory of Bioresource and Applied Microbiology, School of Life Sciences, Henan University, Kaifeng, China
[4] Department of Agroecology, Aarhus University, Tjele, Denmark
[5] iCLIMATE Interdisciplinary Centre for Climate Change, Aarhus University, Roskilde, Denmark
[6] Aarhus University Centre for Circular Bioeconomy, Aarhus University, Tjele, Denmark
[7] State Key Laboratory of Cotton Biology, School of Life Sciences, Henan University, Kaifeng, China, Kaifeng, China
[8] Hanzhong Urban Planning and Architectural Design Institute, Hanzhong, China
* These authors contributed equally to this work.

Corresponding authors
Fengying Liu, fengyingliu@126.com
Dong Wang, wangdong19882005@163.com

## ABSTRACT

Seed germination is a key stage in the life history of plants, which has a crucial effect on plant community structure. Climate change has substantially altered the surface soil temperature and light availability, which can affect seed germination. However, whether the seed germination of different functional groups is affected by the interactions of light and temperature remains unclear. Under laboratory conditions, we examined the effects of low temperature and darkness, as well as their interaction, on the seed germination of 16 species belonging to three plant functional groups (annual and biennials, perennial grasses, and perennial forbs) in a typical steppe, Northern China. We found that low temperature had a significant negative effect on seed germination of all species. Low temperature significantly decreased the final germination percentage and germinative force of the three plant functional groups, and the germination duration of perennial grasses. Darkness significantly decreased the germinative force of perennial forbs and total seeds, and the germination duration of perennial grasses. The interactive effects of light and temperature on the seed final germination percentage and germinative force of perennial grass indicated that darkness strengthened the inhibitory effect of low temperature on the seed germination of the grass functional group. Our study indicate that the seed germination of different plant functional groups varied greatly in response to changing environmental conditions. Our results suggest that future climate change could alter the regeneration and species composition of plant communities through changing seed germination.

## INTRODUCTION

Seed germination is the initial stage of plant growth, and it affects the development and reproduction of individual plants, as well as the structure and composition of the plant community (*Hoyle et al., 2014*; *Zhang et al., 2020*). Compared with the vulnerability of seedlings, seeds are highly tolerant to environmental stress (*Chen et al., 2019*). The establishment of seedlings likely depends on the response of seed germination to the environment. Many plants have dormancy mechanisms to prevent germination, and seeds will break the seed coat and protrude the radicle until the conditions are suitable for seed germination and seedling growth (*Miransari & Smith, 2014*; *Lai et al., 2019*). Germination percentage and germination time determine the timing and location of seedling establishment, and affect species coexistence and plant community development (*Tobe, Zhang & Omasa, 2005*; *Zhang et al., 2020*). There is thus a need to identify the factors that affect the seed germination percentage and germination time.

Temperature and light have critically important effects on seed germination (*El-Keblawy, 2017*; *Chen et al., 2019*). Temperature is essential for breaking seed dormancy and inducing seed germination, as it stimulates enzyme activity in plant seeds, leading to the rupture of the seed coat, and enhances water permeability (*Tabatabaei, 2015*). The response of seed germination to temperature is often characterized by a parabolic relationship (*Chen et al., 2019*). There is an optimal temperature for seed germination, and temperatures above or below the optimum can inhibit seed germination (*Durr et al., 2015*). Ongoing global climate warming has not only resulted in a gradual increase in soil temperature but also led to shorter winters and the melting of snowpack in early spring (*Walck et al., 2011*). The lack of insulation from snow can lead to changes in soil and litter temperature, which can disrupt the regeneration of plants and alter the adaptive ranges of species due to frost exposure (*Walck et al., 2011*). Most studies have focused on clarifying the impact of higher temperatures on seed germination (*Ronnenberg et al., 2007*; *Durr et al., 2015*; *Hadi et al., 2018*). There is thus a need to study the effect of low temperature on seed germination to evaluate the responses of the community composition and structure to climate change.

Light also plays an important role in seed germination (*Baskin & Baskin, 1998*). Light can increase the content and activity of some enzymes in seeds and promote seed germination (*Shahverdi et al., 2019*). Light perceived by plants can be converted into internal signals that result in endogenous phytohormone responses (*Seo et al., 2009*). However, the seed germination of some plants is not light sensitive. For example, the seed germination of *Caragana korshinskii*, which grows on the dunes of Central Asia, shows no response to light (*Zeng et al., 2003*). The response of seeds to light is a mechanism that germination occurs under conditions conducive to seedling growth (*Wang et al., 2014*). Seedlings can be established to meet their own growth and nutritional requirements through photosynthesis (*El-Keblawy, 2017*). In environments where seeds may be buried in deep soil, covered by litter, or sheltered by caregivers, light is an important factor in determining the locations the most suitable for seedling establishment after germination (*Wang et al., 2014*; *El-Keblawy, 2017*). There is thus a need to study the response of

grassland communities to changes in light. Although the effects of temperature and light on seed germination have been extensively studied, how the interaction between low temperature and darkness affects seed germination remains unclear.

The responses of the germination rate of different plant functional groups to nutrients and light vary under global change (*Wang et al., 2014*). Plant functional group is the group of plant species that share key functional traits, have similar response mechanisms to specific environmental factors, and have similar effects on the main ecosystem processes (*Li et al., 2017*; *Su et al., 2018*; *Chen et al., 2020*; *Li et al., 2021*). Therefore, the various responses of seed germination to light and temperature in plant communities may be related to the identity of plant functional groups. In a previous study examining the early succession of Mongolian steppe after drought, the forbs of two *Chenopodium* species had a lower seed germination rate compared with *Salosla collina*, an annual plant (*Kinugasa et al., 2016*). Light can significantly reduce the seed germination rate of perennial grasses regardless of temperature and water conditions (*Hu et al., 2013*). Several studies have investigated seed germination under different environmental factors, but few have examined how light and temperature and their interaction affect the seed germination percentage, germination time, and germinative force of different plant functional groups.

The grasslands in northern China support animal husbandry, yet these grasslands are sensitive to changes in climate and land use patterns (*Zhang et al., 2020*; *Wang et al., 2020a*; *Wang et al., 2021*). There is thus a need to explore the effects of different temperature and light conditions on seed germination of different functional groups. Here, we conducted temperature and light treatment experiments on seeds of typical grassland plants in northern China, and raised the following questions: (1) How do three plant functional groups respond to temperature and light for seed germination, and (2) and whether there was an interaction between temperature and light on seed germination.

## MATERIALS AND METHODS

### Study site and materials

The seeds of this study were collected from a temperate steppe located in Duolun County (42°02′N, 116°17′E, 1,324 m a.s.l), Inner Mongolia, Northern China. The long-term mean annual precipitation of the area is 383 mm, and approximately 90% of the annual precipitation falls during the growing season (May to October). The mean annual air temperature is 2.1 °C. The maximum monthly mean temperature (18.9 °C) occurs in July. January is the coldest month with an average temperature of −17.5 °C. The annual accumulated temperature is 1,600–3,200 °C. The plant community of the grassland ecosystem primarily consists of perennial forbs and grasses; annuals and biennials are also common (*Sagar et al., 2019*; *Miao et al., 2020*; *Wang et al., 2020a*).

The seeds of more than 600 native mature plant individuals from 16 common species were collected in semi-arid grassland from September to October 2017. These species belong to the three main functional groups: perennial forbs (PF), perennial grasses (PG), and annuals and biennials (AB). There were nine PF species (*Artemisia frigida, Taraxacum mongolicum, Potentilla tanacetifolia, Potentilla bifurca, Lespedeza davurica, Medicago ruthenica, Plantago asiatica, Allium tenuissimum* L., and *Thalictrum petaloideum*), four
PG species (*Stipa krylovii, Agropyron cristatum, Pennisetum centrasiaticum,* and *Leymus chinensis*) and three AB species (*Artemisia scoparia, Chamaerhodos erecta,* and *Dontostemon dentatus*) (*Zhong et al., 2019*; *Miao et al., 2020*).

## Seed germination

The seeds were dried and then preserved in the dark at natural temperature until April 2018 for germination experiments. Germination experiments were conducted in 10 cm diameter Petri dishes. This experiment used a factorial design with two factors: light (photoperiod, darkness) and temperature (low temperature, high temperature), which were combined into four different treatments: high temperature/photoperiod (20 °C, 12 h light/12 h dark), low temperature/photoperiod (4 °C, 12 h light/12 h dark), high temperature/darkness (20 °C, 24 h dark), and low temperature/darkness (4 °C, 24 h dark). There were three replicates for each treatment. The different photoperiods were used to simulate the availability of light, darkness is to simulate the expected changes due to nitrogen deposition promotes plant individual growth and litter increase under climate change, which leads to prolongation of dark environment (*Hoyle et al., 2014*; *Chen et al., 2019*). A total of 4 °C was used to simulate the snow-melting field temperature in winter (spring) after seed dispersal, and 20 °C was used to simulate the optimal germination temperature of local seeds (*Hoyle et al., 2014*; *Zhang, 2018*; *Wang et al., 2020b*).

First, 192 Petri dishes (16 species × 4 treatments × 3 replicates) were selected for disinfection, a layer of filter paper was placed in each Petri dish. A total of 20 seeds were evenly distributed in each dish and moistened with a spray bottle. Finally, the Petri dishes were placed in different incubators for the germination experiment. Water was added daily for 60 days to keep the Petri dish filter paper moist. Radicle emergence was used as the criterion for germination, and germinating seeds were immediately removed to reduce the disturbance on other seeds (*Lai et al., 2019*).

## Statistical analysis

Germination was measured using four indices: final germination percentage (FGP), germinative force (GF), germination duration (GD), and germination start (GS):

FGP is the percentage of germinated seeds to tested seeds (*Lai et al., 2019*; *Zhang et al., 2020*); GF is the percentage of seed germination at peak to tested seeds. GF measures the speed and uniformity at which seeds germinate. GF and FGP are the main indexes for measuring the quality of seeds (*Zhou et al., 2020*). GD is the number of days from germination of the first seed to germination of the last seed (*Bu et al., 2008*); GS is the number of days from the start of the experiment to the germination of the first seed (*Chen et al., 2019*).

Using data of the 16 species, generalized linear models (GLM) were used to test the effects of temperature and light and their interaction on seed germination of each plant functional group. The sample sizes of PF, PG and AB in each treatment were 27, 12, 9, respectively. F-tests were conducted to evaluate whether GLM predictors explained a significant fraction of the total deviance or not. Tukey's honestly significant difference (HSD) test was used to evaluate significant differences among multiple treatments based

**Table 1 The effects of light, temperature and plant functional group (PFG) on final germination percentage (FGP), germinative force (GF), germination duration (GD) and germination start (GS) based on generalized linear model analyses.**

|  | FGP | | GF | | GD | | GS | |
|---|---|---|---|---|---|---|---|---|
|  | F | P | F | P | F | P | F | P |
| Light | 1.895 | 0.170 | 4.978 | **0.027** | 1.639 | 0.202 | 1.506 | 0.221 |
| Temperature | 48.237 | **<0.001** | 68.338 | **<0.001** | 4.903 | **0.028** | 378.742 | **<0.001** |
| PFG | 3.698 | **0.027** | 7.837 | **<0.001** | 0.487 | 0.616 | 0.808 | 0.448 |
| Light × Temperature | 1.698 | 0.194 | 2.831 | 0.094 | 0.931 | 0.336 | 0.379 | 0.539 |
| Light × PFG | 0.109 | 0.897 | 0.292 | 0.747 | 2.405 | 0.093 | 0.068 | 0.935 |
| Temperature × PFG | 0.829 | 0.438 | 2.123 | 0.123 | 0.705 | 0.495 | 1.141 | 0.322 |
| Light × Temperature × PFG | 0.962 | 0.384 | 0.375 | 0.688 | 0.954 | 0.387 | 1.777 | 0.172 |

**Note:**
Significant effects ($P < 0.05$) are in bold.

on ANOVA results. Means (±SE) of non-transformed data were calculated and shown in figures. Spearman correlation method was used to determine the correlations among FGP, GF, GD, and GS. All statistical analyses were performed using R software (*R Core Team, 2022*), and the threshold for statistical significance was $P < 0.05$.

# RESULTS

## Seed final germination proportion

Two-way ANOVA indicated that plant functional groups presented a statistically different response in FGP (Table 1, $P < 0.05$). The mean FGP of PG was 22.6%, which was the lowest among the three plant functional groups (Fig. 1). Low temperature significantly inhibited the FGP of total seeds by 29.7% ($P < 0.001$, absolute change, Table 2). Darkness had no significant effect on FPG. There was no interactive effect between temperature and light on the FGP of total seeds. Low temperature significantly inhibited the FGP of PF, PG, and AB by 30.5%, 19.0%, and 41.9%, respectively (Table 2, Fig. 1). The interactions of temperature and light had a significant effect on the FGP of PG ($P = 0.024$). Under photoperiod conditions, low temperature decreased the FGP of PG seeds by 10.0%. Under darkness, low temperature significantly decreased the FGP of PG seeds by 27.9%. Darkness promoted the FGP of PG by 4.2% at high temperature and inhibited the FGP of PG by 13.8% at low temperatures. According to Tukey's honestly significant difference (HSD) test, the values of FGP of perennial forbs and total species were the highest under the high temperature/photoperiod treatment, and were lowest under low temperature/darkness treatment.

## Seed germinative force

There were statistical differences among plant functional groups in FGP (Table 1, Fig. 2). Low temperature and darkness significantly decreased the GF of total seeds by 13.4% and 3.7%, respectively. There was no interaction between the effects of temperature and light on the GF of total seeds. Low temperature significantly decreased the GF of PF, PG, and AB by 11.9%, 8.1%, and 24.7%, respectively (Table 2, Fig. 2). Darkness significantly

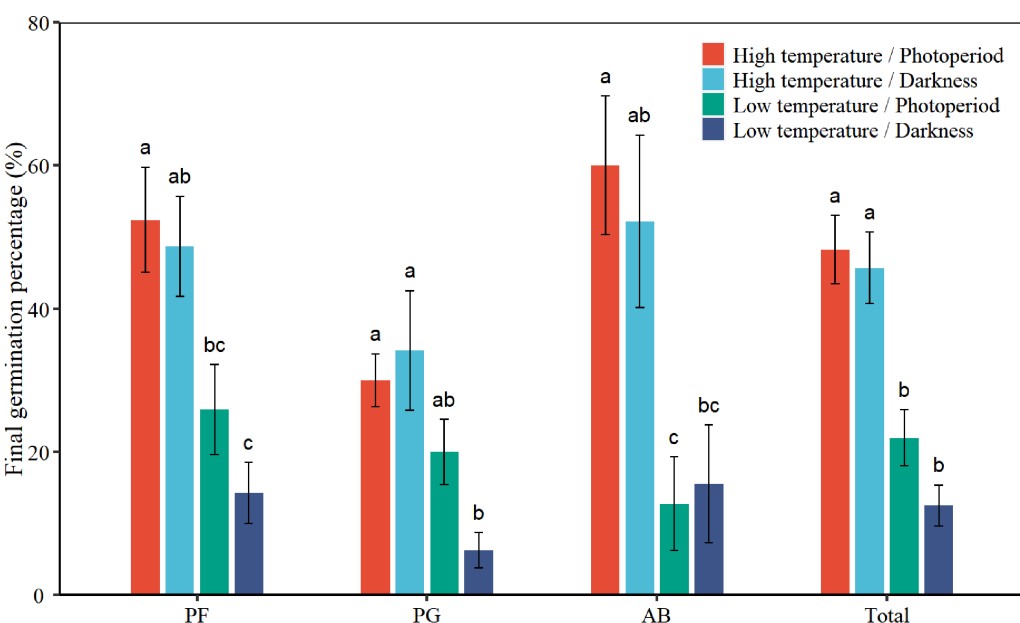

**Figure 1 Effects of temperature (high temperature, low temperature) and light (photoperiod, darkness) on seed final germination percentage of total species (Total), perennial forbs (PF), perennial grasses (PG), and annuals and biennials (AB).** Error bars indicate the standard error of three replicates. The different letters over the bars represent significant difference among the four treatments based on Tukey's honestly significant difference test ($P < 0.05$).

**Table 2 The effects of light and temperature on final germination percentage (FGP), germinative force (GF), germination duration (GD) and germination start (GS) of perennial forbs (PF), perennial grasses (PG), annuals and biennials (AB) and total specie (Total) based on generalized linear model analyses.**

| | | FGP | | GF | | GD | | GS | |
|---|---|---|---|---|---|---|---|---|---|
| | | F | P | F | P | F | P | F | P |
| PF | Light | 1.355 | 0.247 | 3.960 | **0.049** | 0.261 | 0.611 | 0.653 | 0.421 |
| | Temperature | 21.943 | **<0.001** | 24.260 | **<0.001** | 3.616 | 0.060 | 194.280 | **<0.001** |
| | Light × Temperature | 0.879 | 0.351 | 1.734 | 0.191 | 1.277 | 0.261 | 1.621 | 0.206 |
| PG | Light | 0.863 | 0.358 | 0.929 | 0.340 | 10.645 | **0.002** | **0.986** | 0.326 |
| | Temperature | 13.879 | **<0.001** | 30.181 | **<0.001** | 4.047 | **0.050** | **123.118** | **<0.001** |
| | Light × Temperature | 5.042 | **0.030** | 4.080 | **0.050** | 2.235 | 0.142 | 0.154 | 0.697 |
| AB | Light | 0.059 | 0.810 | 0.386 | 0.539 | 0.318 | 0.577 | 0.115 | 0.737 |
| | Temperature | 17.419 | **0.000** | 27.542 | **<0.001** | 0.001 | 0.977 | 67.933 | **<0.001** |
| | Light × Temperature | 0.259 | 0.614 | 0.000 | 0.996 | 0.390 | 0.537 | 1.853 | 0.183 |
| Total | Light | 1.834 | 0.177 | 4.381 | **0.038** | 1.566 | 0.212 | 1.450 | 0.230 |
| | Temperature | 46.694 | **<0.001** | 60.134 | **<0.001** | 4.683 | **0.032** | 364.650 | **<0.001** |
| | Light × Temperature | 1.641 | 0.202 | 2.501 | 0.115 | 0.889 | 0.347 | 0.365 | 0.547 |

**Note:**
Significant effects ($P < 0.05$) are in bold.

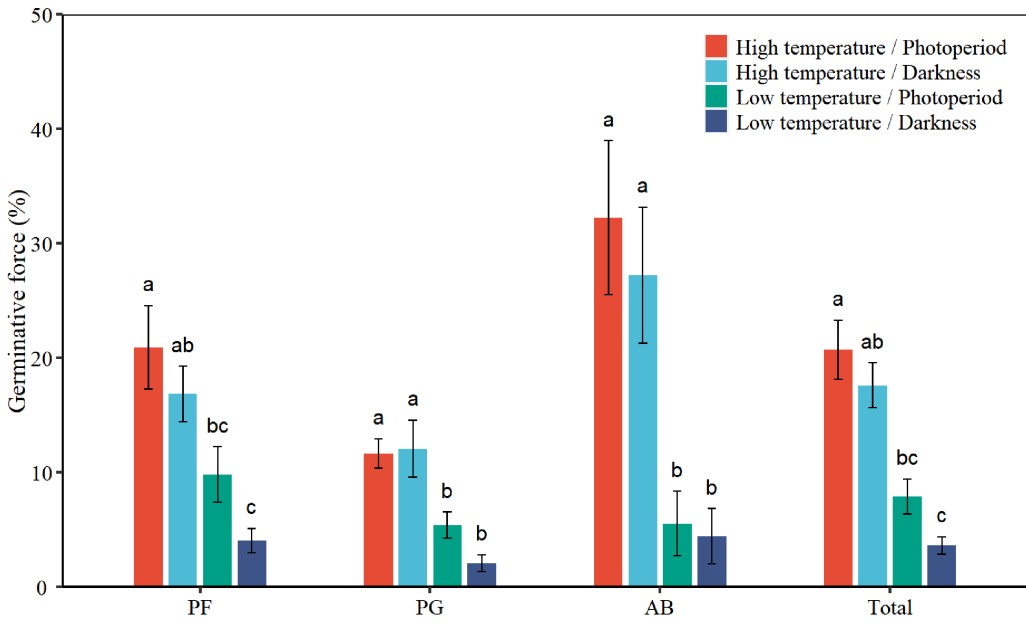

**Figure 2 Effects of temperature (high temperature, low temperature) and light (photoperiod, darkness) on seed germinative force of total species (Total), perennial forbs (PF), perennial grasses (PG), and annuals and biennials (AB).** Error bars indicate the standard error of three replicates. The different letters over the bars represent significant difference among the four treatments based on Tukey's honestly significant difference tests ($P < 0.05$).

reduced the GF of PF by 4.9%. The interaction between temperature and light significantly affected the GF of PG ($P = 0.043$, Table 2). Under photoperiod conditions, low temperature significantly reduced the GF of PG seeds by 6.3%. Under darkness, low temperature significantly decreased the GF of PG seeds by 10%. According to Tukey's honestly significant difference (HSD) test, the values of GF of perennial forbs and total species were the highest under the high temperature/photoperiod treatment, and were lowest under low temperature/darkness treatment.

## Seed germination duration

Low temperature significantly decreased the GD of total seeds by 1.6 days (Table 2, Fig. 3). Low temperature significantly reduced the GD of PG by 2.0 days. Darkness significantly reduced the GD of PG by 3.3 days. There was no interaction effect between temperature and light on the GD of total seeds and the three functional groups.

## Seed germination start

Low temperature significantly prolonged the GS of total seeds by 19.8 days (Table 1, Fig. 4). Low temperature significantly increased the GS of PF, PG, and AB by 18.9, 21.0, and 20.7 days, respectively (Table 2, Fig. 4). Darkness had no significant effect on the GS of seed germination. There was no significant interaction effect between temperature and light on the GS of total seeds and different functional groups.

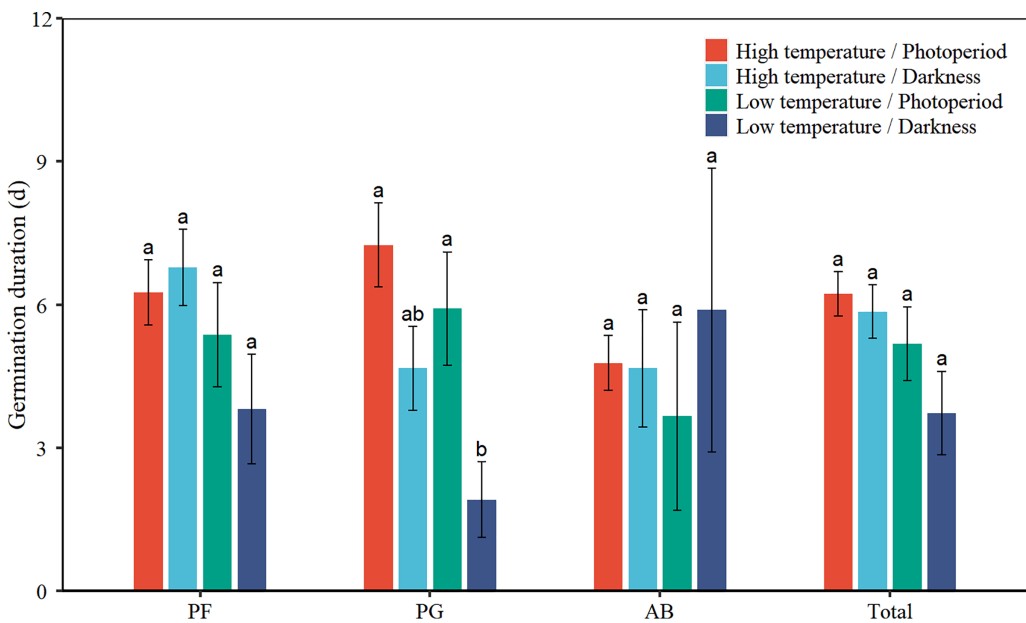

**Figure 3 Effects of temperature (high temperature, low temperature) and light (photoperiod, darkness) on seed germination duration of total species (Total), perennial forbs (PF), perennial grasses (PG), and annuals and biennials (AB).** Error bars indicate the standard error of three replicates. The different letters over the bars represent significant difference among the four treatments based on Tukey's honestly significant difference tests ($P < 0.05$).

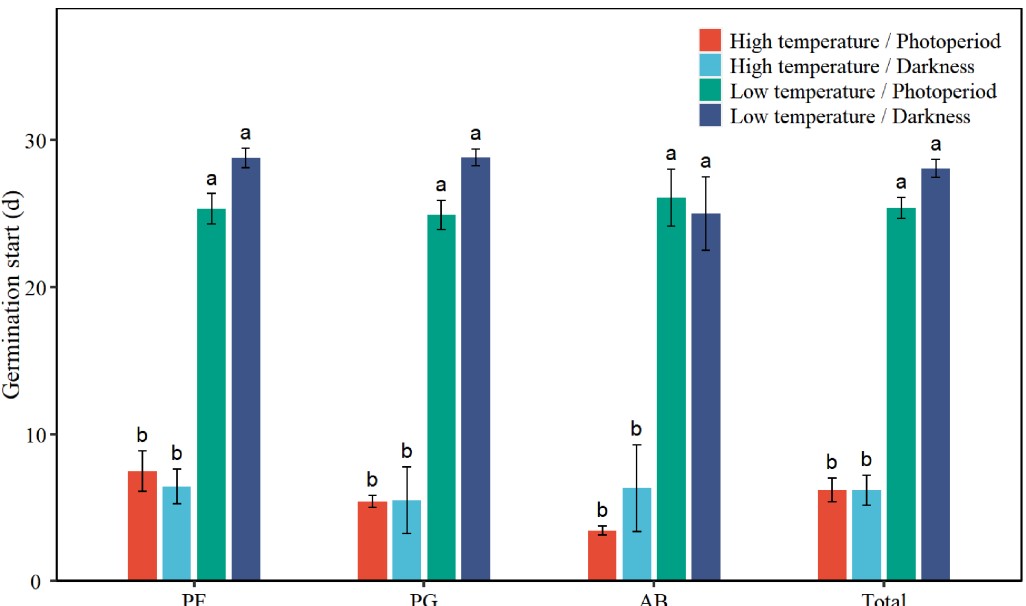

**Figure 4 Effects of temperature (high temperature, low temperature) and light (photoperiod, darkness) on seed germination start of total species (Total), perennial forbs (PF), perennial grasses (PG), and annuals and biennials (AB).** Error bars indicate the standard error of three replicates. The different letters over the bars represent significant difference among four treatments based on Tukey's honestly significant difference tests ($P < 0.05$).

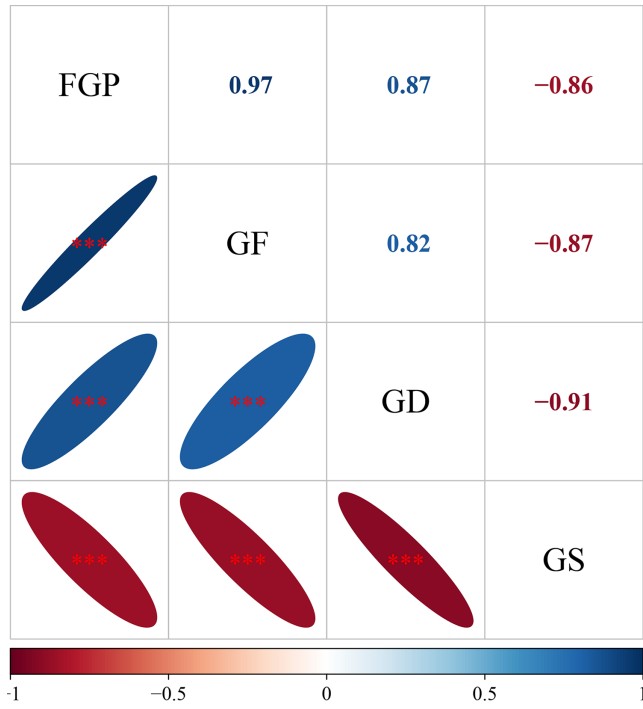

**Figure 5 The relationship among final germination percentage, germinative force, germination duration, and germination start of total species.** Each data point represents the mean value of each species across the four treatments. ***$P < 0.01$, **$P < 0.01$, *$P < 0.05$.

### Relationships between germination indexes

The mean FGP, GF, and GD of all species were negatively correlated with the mean GS (Fig. 5). There was a pairwise positive correlation among the average FGP, GF, and GD of all species.

## DISCUSSION

### Effect of temperature on seed germination

Seed germination was sensitive to environmental conditions, and excessively high or low temperatures were not conducive to seed germination (*Hadi et al., 2018*; *Chen et al., 2019*; *Zhang et al., 2020*). In this study, low temperature significantly decreased the final germination percentage and germination force of all seeds. This is consistent with the results of previous studies indicating that low temperatures could significantly inhibit seed germination (*Lai et al., 2019*). In this experiment, 4 °C was used to simulate the snow-melting field temperature in winter (early spring) after seed dispersal, and 20 °C was used to simulate the optimal germination temperature of local seeds (*Hoyle et al., 2014*). The optimal temperature for seed germination was closely related to the maternal habitat (*Liu, Qi & Shu, 2004*). Suboptimal temperatures could affect the activity of a series of cytoplasmic enzymes and cell membrane permeability, which in turn affected the process of seed germination (*Finch-Savage & Leubner-Metzger, 2006*; *Penfield, 2017*). The low temperature treatment might lead to decreases in the enzyme activity and metabolism in

seeds and thereby inhibit seed germination. Low temperature significantly shortened the germination duration of total seeds. The accumulated cold temperature before seed germination could induce or accelerate seed development and thus shorten the germination duration (*Chen et al., 2019*). Low temperature could also restrict seed germination, as the time required for the germination of the tested seeds increase. Correlation analysis indicated that the duration of seed germination increased as the final germination percentage and germinative force increased (Fig. 5). Decreases in the germination duration indicated that some early germinating species might gain a competitive advantage through increased access to resources (*Wang et al., 2020b*).

Low temperature had different effects on seed germination percentage and germinative force of different functional groups. Compared with perennial forbs and perennial grasses, seeds of annual and biennials were more sensitive to low temperature. This finding was consistent with the results of previous experiments showed that low temperature reduced seed germination of annual plants and induced dormancy (*Zhang et al., 2015*). Short-lived plants had more dormant seeds than long-lived plants as well as more requirements for their seeds to germinate (*Bu et al., 2008*). Compared with perennials, annual plants only produced seeds once in their lifetime and were more dependent on the environment in which seeds germinate. Under harsh environmental conditions, plants had two germination strategies: adventurous germination or dormancy (*Greenberg, Smith & Levey, 2001*). Once an annual plant failed to germinate, it lost the seed genotype that does not germinate, and thus the annual plant goes into dormancy to forego the risk of germination (*Bu et al., 2008*). The final germination percentage was the lowest for perennial grasses. This might be explained by the fact that perennial species did not depend on successful germination in any year, nor on the establishment of a persistent seed bank, because they could survive for a long time through vegetative growth (*Wesche et al., 2006*). Previous study had shown that dominant perennial plants, such as *Agropyron cristatum* and *Stipa gobiaa*, did not produce new seedlings for many years (*Wesche et al., 2006*). Seeds buried in the soil sense temperature changes and selected suitable times to initiate their life cycle (*Chen et al., 2019*). Therefore, short-term changes in the plant community might stem from changes in annual and biennial plants (*Zeng et al., 2016*; *Anniwaer et al., 2020*). In addition, the final germination percentage of total seeds in this experiment was low, this might stem partly from the fact that seeds were stored at room temperature after being collected from the field, which reduced seed vigor (*Liu, Qi & Shu, 2004*; *Shen et al., 2008*). Some studies had shown that seed vigor was better maintained when seeds were refrigerated (*Liu, Qi & Shu, 2004*). The responses of seed germination of perennial and annual plants to low temperature differed, indicating that the various germination strategies employed by different plant functional groups might affect the community structure.

Global warming will likely result in shorter winters and the melting of snow (*Walck et al., 2011*). Reductions in snow cover resulted in colder soil and deeper soil frosts; this could cause germinated seedlings to die or seeds to go back into dormancy, which left more seeds in the soil seed bank (*Walck et al., 2011*). In this study, it was impossible to identify the effect of fluctuating temperatures on seed germination (*Shen et al., 2008*). Seed

of some species could come out of dormancy only after they were exposed to fluctuating temperatures (*Benech-Arnold et al., 2000*). Therefore, it is necessary to further explore the effects of changes in plant seed functional groups on plant community structure under different temperature fluctuations.

## Effect of light on seed germination

Light was a key environmental factor affecting seed germination (*Finch-Savage & Leubner-Metzger, 2006*). After seed maturity and shedding, seeds might be distributed in different environments on the soil surface. For seeds in soil, the spectral composition and irradiance of light were important signals that can indicate the suitability of environmental conditions (*Gu et al., 2005*). Differences in illumination might induce the dormancy or germination of plant seeds (*Gresta et al., 2010*). The dark conditions used in this study had also been examined in previous studies (*Hoyle et al., 2014*; *Chen et al., 2019*). Increased litter, mainly due to nitrogen deposition, limited the availability of light and increased the possibility that plant seeds would be covered when they left the parent plant (*Jensen & Gutekunst, 2003*). Darkness significantly reduced seed germinative force, which might stem from the mechanism by photosensitivity (*Gresta et al., 2010*). The photosensitive properties of plants prevented seeds from being established in shaded environments covered with litter or trees; consequently, appropriate sites needed to be identified to promote the establishment of seedlings after germination (*El-Keblawy, 2017*). Seeds could use light to detect the distance from the ground and thus identified suitable sites to promote the establishment of seedlings after germination (*Flores, Gonzalez-Salvatierra & Jurado, 2016*). Darkness significantly reduced the germinative force of perennial forbs, but had no significant effect on perennial grasses or annual and biennial plants. Previous studies had shown that two *Chenopodium* plants had low seed final germination percentage under the combined action of light and temperature (*Kinugasa et al., 2016*). These differences led to variation in the germination time and space of different species and functional groups in semiarid grassland community. Plant functional groups had evolved different mechanisms to cope with environmental resource scarcity.

The decrease in seed germination under darkness might protect established plant seedlings from limitations in light resources; canopy space was an important factor limiting the establishment of seedlings (*Olff et al., 1994*). The increase in plant litter promoted by nitrogen deposition increased the amount of surface cover and created a dark environment that affected seed germination (*Jensen & Gutekunst, 2003*; *Zhang, Wang & Wan, 2019*). For some plant seeds that are buried under leaf litter, the need for light to induce germination during burial may prevent germination (*Schutz & Rave, 1999*). Therefore, light competition could limit the richness of plant species through seed germination (*Yang et al., 2011*). Under environmental conditions that were not conducive to germination, seeds remain in a dormant state until conditions were suitable (*Hu et al., 2013*). These results indicated that the dark conditions caused by the litter would affect the process of seed germination, and the light limitation of litter could be reduced by proper grazing and mowing in the future to promote plant establishment (*Yuan, Liang & Zhang, 2016*).

## Interaction effect of temperature and light on seed germination

Environmental factors such as temperature and light were key factors affecting seed germination (*Gao et al., 2012*). Seed germination could only respond to specific combinations of environmental factors (*Yi et al., 2019*), and adverse temperature and light conditions, individually or in combination, might prevent the germination of newly shed seeds (*Schutz & Rave, 1999*). In this study, low temperature and darkness had a significant interaction effect on the final germination percentage and germinative force of perennial grass. Darkness intensified the inhibitory effect of low temperature on seed germination of perennial grass. Seed final germination percentage was the lowest under the combined action of darkness and low temperature, and this interaction between light and temperature also affected germinative force of perennial grass (*Wu et al., 2016*). These observations indicated that interactions among different environmental factors could affect seed germination, and differences were observed among the different plant functional groups (*Wu et al., 2016*; *Chen et al., 2019*; *Yi et al., 2019*). Johnson's experiment (2012) showed that the interaction between light and temperature affected seed germination by demonstrating that higher temperatures were required for seeds to germinate in the presence of light (*Johnson & Kane, 2012*). Furthermore, in some plants with strong photosensitivity, seed germination was mediated by temperature-controlled phytochromes (*Yang et al., 1995*). Plants had evolved strategies that involve both predicting germination and optimizing their adaptability, wherein some seeds were allowed to germinate in the current environment while others remain dormant, thus hedging their bets on unpredictable conditions that were not conducive to seedling establishment (*Yi et al., 2019*).

The findings of this study suggested that low temperature significantly inhibited seed final germination percentage, especially that of annual and biennial plants. This effect had also been observed in adult plants in terrestrial ecosystems. Annuals were more sensitive to temperature changes than perennials, and their growth would be promoted by changes in temperature (*Zhou et al., 2011*). Many annual and biennial plants had a better bet-hedging strategy for completing their life cycle earlier under suitable conditions, which provided an advantage in resource competition (*Gremer & Venable, 2014*; *Zhang et al., 2020*). The results of seed germination at the functional group level were consistent with those found at the plant community level, indicating that the response of seed germination to environmental changes could explain community changes. Under multi-factor climate change, the responses of seed germination of the plant community would be complex. Seed germination was a key stage in plant life history, but it was only the first step, and there was still a lot of uncertainty about how the structure of plant community might change. In addition, to verify the long-term effects of climate change on plant community structure, multi-year sampling and increasing sample numbers are required, while focusing on whether seed germination status is consistent with the response of adult plant communities to climate change.

## CONCLUSION

We found that low temperature had significant negative effects on seed final germination percentage, germinative force, germination duration, and germination start at both the community level and the functional group level. The negative effects of low temperature on the final germination percentage and germinative force were higher for annuals and biennials than for other plant functional groups. Perennial grasses were affected by the interaction between low temperature and darkness. Darkness strengthened the inhibitory effect of low temperature on seed final germination percentage and germination force of perennial grasses. The changes in community structure caused by the diverse response of different functional groups affected the original ecological services provided by ecosystems. The responses of seed germination of plant functional groups to changes in the environmental conditions in semiarid grasslands require further exploration for explaining the responses and changes in the ecological function of plant communities under future climate change.

## ACKNOWLEDGEMENTS

Thanks to Duolun Restoration Ecology Station of the Institute of Botany of the Chinese Academy of Sciences for providing the research sites and the support and technical assistance from the people who work there.

### Funding

This work was supported by the Henan Science and Technology Research Project (222102110126), Natural Science Foundation of Henan Province (202300410082), and National Natural Science Foundation of China (NSFC31600380, 31701831). Ji Chen received grants from the EU H2020 Marie Skłodowska-Curie Actions (No. 839806), Aarhus University Research Foundation (AUFF-E-2019-7-1), Danish Independent Research Foundation (1127-00015B), and Nordic Committee of Agriculture and Food Research. The funders had no role in study design, data collection and analysis, decision to publish, or preparation of the manuscript.

### Grant Disclosures

The following grant information was disclosed by the authors:
Henan Science and Technology Research Project: 222102110126.
Natural Science Foundation of Henan Province: 202300410082.
National Natural Science Foundation of China: NSFC31600380, 31701831.
EU H2020 Marie Skłodowska-Curie Actions: 839806.
Aarhus University Research Foundation: AUFF-E-2019-7-1.
Danish Independent Research Foundation: 1127-00015B.
Nordic Committee of Agriculture and Food Research.

## Competing Interests

The authors declare that they have no competing interests.

## Author Contributions

- Mengzhou Liu performed the experiments, analyzed the data, prepared figures and/or tables, and approved the final draft.
- Ning Qiao performed the experiments, analyzed the data, prepared figures and/or tables, and approved the final draft.
- Bing Zhang analyzed the data, authored or reviewed drafts of the article, and approved the final draft.
- Fengying Liu conceived and designed the experiments, prepared figures and/or tables, and approved the final draft.
- Yuan Miao performed the experiments, authored or reviewed drafts of the article, and approved the final draft.
- Ji Chen analyzed the data, authored or reviewed drafts of the article, and approved the final draft.
- Yanfeng Sun performed the experiments, prepared figures and/or tables, authored or reviewed drafts of the article, and approved the final draft.
- Peng Wang performed the experiments, authored or reviewed drafts of the article, and approved the final draft.
- Dong Wang conceived and designed the experiments, authored or reviewed drafts of the article, and approved the final draft.

## Data Availability

The raw data is available as a Supplemental File.

## Supplemental Information

Supplemental information for this article can be found online at http://dx.doi.org/10.7717/peerj.14485#supplemental-information.

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
