# Peer review of "Differential responses of the seed germination of three functional groups to low temperature and darkness in a typical steppe, Northern China"

_PeerJ, doi:10.7717/peerj.14485_

## Round 0.1 · original submission · Major Revisions

Seed germination, as one of the key stages in plant life history, is crucial for plant community structure. In the manuscript, the authors focused on the effects of temperate and light, as well as their interaction on seed germination of sixteen species belonging to three plant functional groups of grassland in northern China. Overall, the manuscript provides considerable useful information that is worth publication. However, there are still some major and minor issues that need to be revised as the reviewers indicated. Below, please find the reviewers' comments for your perusal. Please make corresponding revisions and corrections as suggested by the reviewers.

Reviewer 1 ·

Basic reporting

Seed germination is a key stage in the life history of plants that has a crucial effect on plant community structure. Nowadays climate change has substantially altered the surface soil temperature and light availability, which can affect seed germination. This study examined effects of low temperature and darkness, as well as their interaction, on the seed germination of sixteen species belonging to three plant functional groups (annual and biennials, perennial grasses, and perennial forbs) in a temperate steppe, Northern China, hoped to discover how the seed germination of different functional groups is affected by the interactions of light and temperature, and found that differential responses of the seed germination of three functional groups to low temperature and darkness.

Experimental design

1. Seed germination is a key stage in the life history of plants, but only the first step. There are only very a few germinated seeds can survive and develop into adult plants. So it is too simple to infer changes in community structure from changes in seed germination;
2. In this study there are not enough species included, only 16 species (9 perennial forbs, 4 perennial grasses and 3 annuals and biennials);
3. It seems that only a few seeds were used in this study. In Line 119, the authors said that more than 600 local individual seeds were collected from 16 species; well in Line 135-136, they said that there were 192 Petri dishes, each contained 20 seeds. How did this happen?
4. I think that the temperature treatments in this study was not reasonably designed, because the study area has an average temperature of -17.5°C in the coldest month (January), and 18.9°C in the warmest month (July). Climate changes only shorten the period of time suitable for seed germination, which do not mean a change of the average temperature in the warmest from 20°C to 4°C;

Validity of the findings

5. Different species vary in seed viability and germination time, but these are common response of seed germination, including decreased germination percentage and prolonged germination time in suboptimal temperature for most seeds, and decreased germination percentage and prolonged germination time in darkness for small seeds. In this study the germination test lasted only 60 days, furthermore the ungerminated seeds were not checked when the test finished. We do not know if these ungerminated seeds dead or viable? Would they germinate later? Did the germination profile change?

Additional comments

6. The discussion section is too long but not to the key: Why seeds of these plant functional groups (perennial forbs, perennial grasses, and annuals and biennials) have such different responses?

Reviewer 3 ·

Basic reporting

Seed germination has been considered as a key plant life history stage, which is crucial for population regeneration and community structure. In this manuscript, authors addressed the research question that whether the seed germination of different functional groups is affected by low temperature and darkness as well as their interaction, which is meaningful to explain and predict plant community composition and structure in grassland ecosystems under climate changes. Overall, this manuscript is well-organized and well-written. Therefore, I suggest this manuscript can be accepted after some minor revisions.

Experimental design

The experimental design is reasonable, and the tables and figures are relevant and with high quality.

Validity of the findings

The results are reliable and can support their conclusion. The conclusions are well stated and closely linked to the research questions authors proposed in the Introduction section.

Additional comments

Some specific comments are as follows:
1.In the title, I suggest authors to use “typical steppe” rather than “typical grassland”
2.In the Abstract section, I suggest some results can be merged, e.g., L27-28 and L29-30. Besides, I suggest these two sentences in L36-39 should be incorporated as the final conclusion, for example, “Our study indicate that the seed germination of ...., future climate change could alter population regeneration and community composition in grassland ecosystems through its effects on seed germination”
L44: “as well as the structure and composition....”
L51-52: this sentence needs to be rephrased: “...of seedling establishment, and affect species coexistence and plant community development”
L68: I suggest to change into “how community composition and structure will be...”
L75: “the responses of seeds to light”
L80-83: I suggest this sentence should be deleted or rephrased
L106-107: I think this last sentence should be rephrased
L132-133: For the high temperature/darkness treatment, I think there might be a mistake, it should be 20 ℃, 24 h dark, please correct it
L221: I think here it should be “increased” rather than “reduced”
L239: I think the “persistent seed bank” might be better
L258-259: This sentence seems to be confused, please rephrased it
L262: “seeds”, not “seedlings”

·

Basic reporting

The paper is very well writen, it is clear the background and the need to understand the interaction effect of temperature (low-high) and light (photoperiod and darkness) on seed germination. The ideas are clearly expressed and the structure is coherent. References are consistent with the topic investigated in the paper.

I would suggest some chenges in table 1 and table 2 to improve the clarity of the results obtained (see 3. Validity of the findings).

I strongly recomend to check the statistical analysis and therefore, the results presented. In the following sections I explain it in more detail.

Experimental design

I consider the experimental design (low – high temperature and light – darkness) provides valuable information on the impact of climate change on seed germination. However, I have some remarks on the design.

My first question is about the selection of the species per functional group and the number of species per group. You have the following setting:
Perennial forbs (9 species)
Perennial grasses (4 species)
Annual and biennials (3 species)

Why the differences in number of species per functional group? Having a more similar number of species per group would improve the statistical power. The species mentioned in annual and biennials (A. scoparia, C. erecta and D. dentatus) are forbs. Why you did not include grasses as well?

I recommend giving more information about how did you select the species per functional group and why did you select forbs in annual and biennials and not grasses (e.g. due to species composition, abundance, importance of the species for human use, etc).

Seed germination

Please, correct in line 133: darkness (20 °C, 12 h light / 12 h dark) for (20 °C, 24 h dark)

It is not clear why did you choose those two temperature values and photoperiods. You mention in line Line 134-135 "The different photoperiods were used to simulate the availability of light". But the darkness is to simulate the expected changes due to climate change? The selected value of high temperature is 20 °C, which is higher than the mean monthly temperature (18.9 °C) that you report in line 114-115. But 4 °C is higher than the mean annual air temperature (line 113-114). It would be better to explain directly in the text why you selected those values of temperature (e.g. higher, lower than current temperature of growing season) and photoperiod (e.g simulate variation due to climate change; prolonged darkness due variation). In this way, other researchers can clearly understand the criteria for your experimental design and replicate them. Otherwise, readers can make different assumtions.

Statistical analysis

In the statistical analysis starting in line 154 you state “Generalized linear models were used to test the effects of temperature and light on the seed germination of each plant functional group”. But later in line 157 you said that “Tukey´s honestly significant difference (HSD) test was used to evaluate significant differences among multiple treatments”.

First, I recommend to clarify in the text if you did GLM or Anova (nested) (Binomial Anova is the same as GLM binomial) and please mention the structure of the model to understand if you include the interaction term or not. Please also check in the text the phrase of the Tukey test, because if you did a GLM it is not possible to do a Tukey test, it only works for Anova. Beware that if you try to do a Tukey test for a GLM in R you get an error message.

In line 159 and 160 you mention that “A linear model (y= a + bx) was used to determine the pairwise relationships among FGP, GF, GD, and GS”, however, it is not clear why you used a linear model instead of correlation. To use a linear model, you must know the biological relationship between the response variable and the explanatory variable. Which of your variables of seed germination (FGP, GF, GD, GS) would be the response variable and which the explanatory? I would suggest to replace this analysis for a correlation analysis, because all variables relate to seed germination respond to intrinsic and extrinsic factors and you want to know only the relationship among them (correlation has not directionality). A linear model assumes that changes in one variable would explain changes in the response variable. Please, check these assumptions and the analysis for this section as well. If you keep the linear model, please, include the biological background to assume which are the response variables and which the explanatory variables.

Providing some extra information and clarifying the issues mentioned above, will improve the understanding of the findings reported in your research.

Validity of the findings

In Table 1. and Table 2., it is reported the factor "light" apart from "temperature". How could you disentangle light from temperature if the experimental setting has 2 factors for darkness: Low temperature (4C) and high temperature (20C). Could you please explain this in the results? and if is needed, in methods.

It is not clear if in table 1 you report the output of GLM for all species and in table 2 you report these results per functional group. If so, please write it in the thext of both tables. Moreover, in methods was mentioned that for FGP and GF the Binomial model was used, and for the variables GD and GS a Poisson model was applied. Please, add also this information to the table. When reporting a GLM output it is important to include more information from the output, from which the readears can infer the positive or negative relationship and the significance, thus, I would recomend to add in Table 1 and 2 the information about the Estimate, Standard Error, F or Z value (p-value is reported) obtained in the analysis.

When running your data in R, I do obtain the terms like this (just to show few lines):

PF
PG
treatmentHigh temperature / Photoperiod
treatmentLow temperature / Darkness
PF – treatmentHigh temperature / Photoperiod
PG – treatmentHigh temperature / Photoperiod

From this output, I did not obtain the same p-value as reported in Table 1 neither Table 2. Additionally, I could not separate the effect of Darkness from Temperature from the data set provided.

When reporting the correlation among seed germination factors in line 201. is wirtten "There was a pairwise positive correlation among the average FGP, GR and GD of each species", however figure 5. is showing a negative (e.g. FGP and GS) and positive relationship (e,g GF and GD) among these variables. Please check this phrase based on what is showed in the graph (Perhaps you wanted to say that all of them are significant).

Therefore, I strongly recommend to check again the statistical analysis:
- Clarify if you did a Binomial – Poisson GLM analysis or a nested Anova and a post-hoc Tuckey test (and report the later in the results)
- Check the output of these analysis and explain how did you obtained separately the effect of Darkness from Temperature
- Check the lineal model and correlation

It is important to check first all these issues, it would be necessary to write again some results (Tables and the text) and later check again if the discussion needs also some changes. The general scope of the paper is very interesting, but the reported results need some improvement.

Additional comments

Line 103. I would suggest to change the phrase "and light conditions on the seed germination" to "and light conditions on seed germination"

Line 104. I would suggest to change the phrase "the response of the seed germination of three plant" to "the response of seed germination of three plant"

Line 106 and 107. "The results of this research provide important data that will aid the conservation of species diversity under climate change". Could you develop this idea and mention how.

Line 132-133. When you describe the experimental setting: “high temperature / darkness (20 °C, 12 h light / 12 h dark),” would be corrected to (20 °C, 24 h dark)

Line 166. In the phrase "The mean of FGP of FG was 22.6%", the functional group FG must be PF or PG. Pleas chek this.

---

## Round 0.2 · Minor Revisions

Please revise or explain the issues raised by the reviewers.

·

Basic reporting

no comment

Experimental design

no comment

Validity of the findings

no comment

Additional comments

no comment

Reviewer 3 ·

Basic reporting

The manuscripts “Differential responses of the seed germination of three functional groups to low temperature and darkness in a typical steppe, Northern China” demonstrates the interaction effect of temperature (low-high) and light (photoperiod and darkness) on seed germination of different functional groups. The ideas are clearly expressed and the structure is coherent. The structure is well-organized and the ideas are clearly expressed. However, there are some minor issues, which need to be revised before considering for publication. I recommend paying attention to the tenses of the writing to ensure a better expression.

Experimental design

The experimental design is reasonable and provides valuable information to explain and predict plant community composition and structure in grassland ecosystems under climate changes.

Validity of the findings

The results were reliable and supported their conclusions.

Additional comments

Some suggestions are as follow:
L24 Using a laboratory or field experiment?
L33-36 This sentence is far too long.
L64 change “result” to “resulted”
L116 “1324 m a.s.l” not “1324 m.a.s.l”
L132 change “Allium tenuissimum L.” to “Allium tenuissimum L.”
L244 change “indicate” to “indicated”
L249 change “showing” to “showed”
L253 change “perennial” to “perennials”
L259 change “percentage was lowest for perennial grasses” to “percentage was the lowest for perennial grasses”

·

Basic reporting

The paper is clearly written. The introduction is well organized as well as the topics in the discussion. The structure in each section lets the reader understand the topics and follow the complete history presented in the paper.

The literature cited in the paper support very well the introduction and the findings reported by the authors.

Experimental design

The experimental design and the statistical analysis are consistent with the aims of the study. It is very interesting to include darkness as a factor that may influence seed performance in future scenarios of climate change. Changes made in Methods clarify the experimental set up and the analysis done in the study. I would only suggest to specify the sample size used in the generalized linear models for each functional group and for each index (FGP, GF, GD, GS). Sample size constitutes a key parameter in statistics, because it determines the statistical power. Large sample sizes may contribute to avoid Error Type I.

Validity of the findings

I consider that the changes made in general had improved the understanding of the findings and the data analyzed. Specifically, changes made in Table 1 enhance the interpretation of the results, however, Table 1 is only presented in the document “Response to comments”, and Table 2 header´s is also presented here, but were not updated in the manuscript neither in the files. Please, check this and update the files. In Table 1 you report a F-value, however, when doing a glm with the error distribution poisson or binomial, the output gives a Z-value. Please, also check this information and correct in the text if needed.

Checking Table 2, seems that it represents the results of the Tukey test because it shows the source of variation for each variable. Please, check this and if so, please specify it in the title. Otherwise, the results from the Tukey test are not mentioned in the manuscript.

Additional comments

Please, check:

In Line 38. "Indicated the seed germination of the seed germination of different plant" is double written the seed germination

In Line 40. conditions,future climate change could alter the regeneration and species. There is a double space.

In Line 142. germination.MATERIALS AND METHODS. The title of MTERIALS AND METHODS is in the same line.

In Line 144 to 172 the font and its distribution is different from the rest of the text.


In Line 228 and 237. Two-way ANOVA indicated that FGP was significantly affected by plant functional group (Table 1, P < 0.05, and Line 249 and 250. Plant functional group had significant effects on GF, the mean GF of AB was the highest and PG was the lowest. I would suggest to rephrase these; plant functional groups present a statistically different response in FGP and GF. It is not a factor that you are manipulating (as light or darkness), but it is a groping factor that you expect to respond in a different way, as you explained it in the introduction.

In Line 362 I would suggest replace nitrogen availability by nitrogen deposition

---

## Round 0.3 · accepted · Accept

The paper is well organized, and all the concerns and issues raised by the reviewers have been taken care of.